# Catalytic Etherification of *ortho*-Phosphoric Acid for the Synthesis of Polyurethane Ionomer Films

**DOI:** 10.3390/polym14163295

**Published:** 2022-08-12

**Authors:** Ilsiya M. Davletbaeva, Oleg O. Sazonov, Ilyas N. Zakirov, Ruslan S. Davletbaev, Sergey V. Efimov, Vladimir V. Klochkov

**Affiliations:** 1Technology of Synthetic Rubber Department, Kazan National Research Technological University, 68 Karl Marx str., Kazan 420015, Russia; 2Material Science and Technology of Materials Department, Kazan State Power Engineering University, 51 Krasnoselskaya str., Kazan 420066, Russia; 3Institute of Physics, Kazan Federal University, 18 Kremlevskaya str., Kazan 420008, Russia

**Keywords:** etherification of *ortho*-phosphoric acid, catalytic activity of tertiary amines, ethers of *ortho*-phosphoric acid, ionomers films

## Abstract

The etherification reaction of *ortho*-phosphoric acid (OPA) with polyoxypropylene glycol in the presence of tertiary amines was studied. The reaction conditions promoting the catalytic activity of triethanolamine (TEOA) and triethylamine (TEA) in the low-temperature etherification of OPA were established. The catalytic activity of TEOA and TEA in the etherification reaction of phosphoric acid is explained by the hydrophobic-hydrophilic interactions of TEA with PPG, leading, as a result of collective interactions, to a specific orientation of polyoxypropylene chains around the tertiary amine. When using triethylamine, complete etherification of OPA occurs, accompanied by the formation of branched OPA ethers terminated by hydroxyl groups and even the formation of polyphosphate structures. When triethanolamine is used as a catalyst, incomplete etherification of OPA with polyoxypropylene glycol occurs and as a result, part of the phosphate anions remain unreacted in the composition of the resulting aminoethers of *ortho*-phosphoric acid (AEPA). In this case, the hydroxyl groups of triethanolamine are completely involved in the OPA etherification reaction, but the catalytic activity of the tertiary amine weakens due to a decrease in its availability in the branched structure of AEPA. The kinetics of the etherification reaction of OPA by polyoxypropylene glycol catalyzed by TEOA and TEA were studied. It was shown that triethanolamine occupies a central position in the AEPA structure. The physico-mechanical and thermomechanical properties of polyurethane ionomer films obtained on the basis of AEPA synthesized in a wide temperature range were studied.

## 1. Introduction

The synthesis of ionomers, which are polymeric materials with a small number of ionic groups, is a promising way to create new functional materials. Functional ionic groups are used to influence such properties of polymers as the glass transition temperature, size and distribution of domain supramolecular structures in the continuous polymer phase, miscibility of polymers, rheological characteristics, and to improve the mechanical properties of polymers [1,2,3,4,5,6,7,8,9,10]. The ionic fragments are generally incorporated into these polymers as cationic, anionic, or zwitterionic particles in various concentrations. The classification of ionomers depends on how the ionic groups are incorporated into the polymer structure. For example, polyelectrolytes contain ionic groups covalently bonded to the main polymer chain but have a much higher level of molar substitution of ionic groups (usually more than 80 mol%) [3]. Unlike polyelectrolytes, ionomers contain a small number of ionic groups (up to 10–15 mol%). Depending on the type and nature of polymers, ionic groups can be located along the main chain periodically, randomly, or in the form of end groups. These two classes of polymers containing ionic groups have significantly different morphological and physical properties [11]. Ionic groups and polymer chain fragments containing them can combine into nanodomains. Since ion pairs are attached to the polymer chain by covalent bonding, ionic nanodomains behave like multifunctional nodes of the physical spatial polymer network and significantly affect the mechanical properties of polymer materials. Nanodomains 3–4 nm in diameter can contain up to 10–30 ion pairs [11,12].

In the work [13], it was shown that the change in the properties of ionomers in the solid state is due to the clustering of ion pairs in a medium with a low permittivity. The presence of ionogenic groups in the polymer matrix leads to the emergence of specific interactions between macrochains due to ion-dipole or complex-forming interactions [5,14]. By changing the nature of ionic groups, it becomes possible to exert a significant influence on such interactions and, as a result, on the morphology and properties of polymers [15]. In works [16,17,18,19,20,21,22,23,24,25], morphological models of ionic clusters were obtained.

Polyurethane ionomers are of practical and scientific interest in this direction. Polyurethane ionomers (PUIs) combine the advantages of both polyurethanes and ionomers. For example, conventional PUs are very often hydrophobic in nature [26,27,28,29,30], but it is possible to include ionic hydrophilic segments in the PU chain, opening up the possibility of their dispersion and emulsification [31,32].

Polyurethane ionomers can be synthesized in a variety of ways, including using ionic diols or post-ionization of polyurethanes. The review [33] discusses the synthesis, structure-property relationship, and the broader impact of ionic segmented polyurethanes on various fields of research. It was concluded that incorporating biodegradability into already biocompatible systems allows ionic polyurethanes to play an important role in many biomedical applications, including regenerative medicine and antimicrobials. Excellent film and coating properties, combined with technically conductive membranes, produce polymer electrolytes, and end group functionalization provides photo-crosslinkable water-based PU coatings.

In [34,35], polyurethane cationomers were synthesized using an ionic liquid (IL) based on imidazolium diol as a chain extender. Compared to the non-ionic PU analogue, imidazole-containing PU showed reduced hard segment crystallinity and hydrogen bonding due to disruption of chain regularity and ionic association. Improved microphase separation resulted in a higher modulus of the polyurethane containing imidazolium. It was shown that ILs are predominantly located in ionic hard domains. As a result, the ionic conductivity of polyurethane membranes containing imidazolium increased by five orders of magnitude. In [36], phosphonium-containing polyurethanes were synthesized and studied. In polyurethane ionomers, ionic interactions predominated, and the presence of large ionic aggregates was confirmed. Polyurethanes showed excellent elasticity (>1100%) and high tensile strength (>19 MPa).

The use of ionogenic diols in reactions with diisocyanates is described in [37,38,39,40,41,42,43,44,45,46] and is a common method for preparing ionomers. The most commonly used are sulfonic [47,48], phosphoric, and carboxylic acids [49,50,51].

Promising polyol components for the production of polyurethane ionomers are ethers of *ortho*-phosphoric acid (OPA). Ethers of phosphoric acid are used mainly as natural phospholipids. In [52], it was shown that phospholipid fibers obtained by electrospinning make it possible to directly fabricate biological membranes with a large surface area without the use of several synthesis steps, complex electrospinning designs, or surface post-treatment.

The most widely used phosphorylating agent for alcohols is phosphorus oxychloride, which is characterized by high toxicity.

The direct use of accessible and low-toxic OPA for the production of ethers of *ortho*-phosphoric acid is limited due to its low reactivity with respect to hydroxyl-containing compounds. In [53,54], the reaction of alcohols with phosphoric acid in the presence of various tertiary amines was studied. The reaction in the absence of a tertiary amine gives the phosphate monoether in an 8% yield. When using more than 1 eq., Bu_3_N conversion of phosphoric acid reaches a plateau at 55–58%. In addition, it was found that the conversion of OPA depends on the structure of the tertiary amine. The problem with developing such an approach to synthesis was that tertiary amines and phosphoric acid form ammonium phosphates. That is, the use of tertiary amines in the OPA etherification reaction is based on the preservation of their catalytic activity by eliminating the possibility of the formation of ammonium phosphates.

In [55,56,57,58,59], aminoethers of *ortho*-phosphoric acid (AEPA) were synthesized by reacting triethanolamine (TEOA), OPA, and polyoxypropylene glycol (PPG). We used such a relative content of the initial reagents when all three hydroxyl groups of triethanolamine and one hydroxyl group of PPG could potentially be consumed for the etherification reaction. The main branching center of AEPA is a tertiary amine, and the function of subsequent branching centers is performed by phosphates (Figure 1a).

When triethylamine is used, complete etherification of OPA occurs, followed by the formation of branched ethers of *ortho*-phosphoric acid (EPA) terminated by hydroxyl groups. An increase in the OPA content leads to the formation of total phosphates (Figure 1b).

On the basis of AEPA and polyisocyanates, organophosphorus polyurethane ionomers (PPUI) were first obtained, a distinctive feature of which was the ability to combine phosphate groups into clusters (Figure 1). It was found that clustering in polyurethane ionomers is due to the peculiarities of the chemical and supramolecular structure of AEPA. The obtained PPUIs were investigated as water vapor permeable and pervaporation membranes [57] and as a basis for the development of gel-polymer electrolytes for lithium power sources [58]. 

In [55], it was found that polyurethane ionomers obtained on the basis of AEPA, in the formation of the central unit of which triethanolamine participates, exhibit significantly higher mechanical properties compared to polyurethanes obtained on the basis of EPA.

However, in previously published materials, the mechanism of the catalytic activity of tertiary amines (TA) in the OPA etherification reaction was not established. In connection with the above, the aim of this work is to study the reaction conditions that promote the catalytic activity of tertiary amines in the reaction of low-temperature OPA etherification. The influence of the chemical structure of tertiary amines on the completeness of the etherification reaction of OPA with polyoxypropylene glycol was studied. As tertiary amines, triethanolamine, capable of participating in the etherification reaction, and triethylamine (TEA), not containing hydroxyl groups in its composition were used. On the basis of aminoethers of *ortho*-phosphoric acid, synthesized under different reaction conditions, polyurethane ionomer films have been obtained and studied.

## 2. Materials and Methods

### 2.1. Materials

Polypropylene glycol, MW = 1000 (PPG, Wanol 2310), was purchased from Wanhua Chemical (Beijing, China). Triethanolamine (TEOA), triethylamine (TEA), and an 85% aqueous solution of *ortho*-phosphoric acid (H_3_PO_4_, OPA) were purchased from Ltd «MCD-Chemicals» (Moscow, Russia). Polyisocyanate “Wannate PM-200” (PIC) was purchased from (Kumho Mitsui Chemicals, Inc., Beijing, China).

### 2.2. Synthetic Procedures: General Procedure for Synthesis of Ethers/Aminoethers of Ortho-Phosphoric Acid (EPA/AEPA)

To obtain AEPA, triethanolamine, *ortho*-phosphoric acid, and PPG were used at molar ratios of [TEOA]:[H_3_PO_4_]:[PPG] = 1:0.5 ÷ 6:6 (AEPA-2 ÷ 6). At the first stage, the calculated amount of H_3_PO_4_ and PPG were placed in a round-bottomed flask, and the reaction mass was stirred for two hours at T = 90 °C with a residual pressure of 0.7 kPa to remove residual moisture for two hours. At the second stage, the calculated amount of triethanolamine was added to the flask and the reaction mass was stirred for two hours at T = 90 °C with a residual pressure of 0.7 kPa. The synthesized liquid AEPA was collected in a stoppered flask. The amount of residual water did not exceed 0.3 wt%.

For studies of AEPA-6 obtained at synthesis temperatures (T_synth_) of 30, 40, 50, 60, 70, 80, 90, 100, and 110 °C, the synthesis was carried out similarly, with the only difference that the corresponding synthesis temperature was maintained at the second stage.

The use of triethylamine as a model compound for studying the mechanism of the catalytic activity of tertiary amines in the OPA etherification reaction is due to the greatest similarity of its chemical structure with triethanolamine. The etherification of H_3_PO_4_ with TEA and PPG was carried out similarly to the synthesis of AEPA. Compounds were obtained at molar ratios of [TEA]:[H_3_PO_4_]:[PPG] = 1:(2÷6):6 (EPA-2÷6). The reaction progress was monitored by titration to determine the hydroxyl group concentration.

### 2.3. General Procedure for the Synthesis of Polyurethane Ionomer Films Based on Aminoethers of Ortho-Phosphoric Acid 

The synthesized AEPA-6 (1 g) was mixed with PIC (1 g), stirred for 5 min at room temperature, and cast onto prepared surfaces. The curing of polyurethanes (AEPA-6-PU) was carried out for 24 h at room temperature. 

### 2.4. Measurements

#### 2.4.1. Measurements of the Surface Tension

The droplet counting method was used to determine the surface tension (σ). The basis of the calculations is the law, where the weight of the drop that comes off the pipette is proportional to the surface tension of the fluid and the radius of the pipette (R): m = 2π·R·σ/g, where: g is the acceleration of gravity; m is the drop mass of the test liquid.

#### 2.4.2. The Dynamic Viscosity and Density Measurements

The dynamic viscosity of samples was determined at a temperature of 20 °C at atmospheric pressure and using an SVM 3000 Stabinger Viscometer (Anton Paar, Graz, Austria), with an error of 0.00005 mPa∙s. At the same time, the density of the samples was determined with an error of 0.0005 g/cm^3^.

#### 2.4.3. Light-Scattering of Solution

Dynamic light scattering experiments were carried out on Zetasizer Nano ZS (Malvern, Worcestershire, Great Britain). This instrument has a 4 mW He–Ne laser, which works at a 632.8 nm wavelength. Measurements were carried out at a 173° detection angle. The experiments were carried out at 25 °C in disposable plastic cuvettes of 1 cm path length.

#### 2.4.4. NMR Spectroscopy

^1^H NMR spectra were obtained on a Bruker Avance II 500 spectrometer (Bruker, Rheinstetten, Germany) (500.13 MHz for ^1^H) using a direct BBO probehead (BB-1H-2D). The samples contained about 200 μL of the investigated polymer mix and 400 μL of deuterated chloroformin standard 5 mm NMR tubes. The spectra were recorded at 30 °C. The proton chemical shift scale is referenced with respect to the residual solvent signal.

^31^P NMR spectra of the same samples were obtained on a Bruker Avance II 500 spectrometer (202.46 MHz for ^31^P) using the BBO probehead (BB-1H-2D). 

#### 2.4.5. Water Concentration Measurement

Water concentration was measured on a Mettler Toledo V20 (Mettler Toledo, Zurich, Switzerland) volumetric titrator according to Karl Fischer.

#### 2.4.6. Fourier-Transform Infrared Spectroscopy Analysis (FTIR)

The FTIR spectra of the products were recorded on an InfraLUM FT 08 Fourier-transform spectrometer (Lumex, St. Petersburg, Russia) using the attenuated total reflection technique. The spectral resolution was 4 cm^–1^, and the number of scans was 32.

#### 2.4.7. Determination of The Content of Ortho-Phosphoric Acid by Titration

The mass fraction of phosphoric acid was determined by titration of POH groups with a standard solution of sodium hydroxide in the presence of phenolphthalein.

#### 2.4.8. Mechanical Loss Tangent Measurements (MLT)

The MLT curves of polymer samples were taken using the dynamic mechanical analyzer DMA 242 (Netzsch, Selb, Germany) in the mode of oscillating load. Force and stress–stain correspondence were calibrated using a standard mass. The thickness of the sample was 2 mm. Viscoelastic properties were measured under nitrogen. The samples were heated from –50 °C to 350 °C at a rate of 3 °C/min and a frequency of 1 Hz. The mechanical loss tangent was defined as the ratio of the viscosity modulus G″ to the elasticity modulus G′.

#### 2.4.9. Thermomechanical Analysis (TMA)

The thermomechanical curves of polymer samples were obtained using the TMA 402 F (Netzsch, Selb, Germany) thermomechanical analyzer in the compression mode. The sample thickness was 2 mm, and the rate of heating was 3 °C/min from –50 °C to 350 °C in the static mode. The load was 2 N.

#### 2.4.10. Tensile Stress-Strain Measurements

Tensile stress-strain measurements were obtained from the film samples of size 45 mm × 15 mm with the Universal Testing Machine Inspekt mini (Hegewald&PeschkeMeß und Prüftechnik GmbH, Nossen, Germany) at 293 ± 2 K, 1 kN. The crosshead speed was set at 50 mm/min and the test continued until sample failure. A minimum of five tests were analyzed for each sample, and the average values were reported.

## 3. Results

### 3.1. Ortho-Phosphoric Acid Etherification

In connection with the established regularities in the formation of the AEPA and EPA structures, it seemed necessary to study the catalytic activity of tertiary amines in the OPA etherification reaction. The starting point of the research was the fact that OPA with TEOA and TEA forms tertiary ammonium phosphates, which are not able to exhibit the properties of catalysts for the OPA etherification reaction. Since the reaction system consists of only three components, two of which interact with the formation of tertiary ammonium phosphates, one would expect the key role of polyoxypropylene glycol in the formation of catalytically active centers during the interaction of PPG with TEOA and TEA.

Some physicochemical properties of triethylamine solutions in PPG were studied. Since the boiling point of TEA is 90 °C, under the conditions of AEPA and EPA synthesis (T = 90 °C, residual pressure 0.7 kPa), triethylamine could either fly away or remain in the reaction system in the form of tertiary ammonium phosphate, losing at this point its catalytic activity.

In order to demonstrate the ability of PPG to enter into intermolecular interactions with TEA, solutions of TEA were prepared in a number of oligoetherdiols potentially capable of etherification of *ortho*-phosphoric acid. Solutions of TEA in polyol Puranol 373, polyoxytetramethylene glycol, propylene glycol, and PPG in equimolar amounts were placed in a flask and heated at T = 90 °C, a residual pressure of 0.7 kPa, and mixed in within 2 h. The TEA solution in the listed glycols, together with the flask and stirrer, was weighed before and after heating at the residual vacuum pressure. The residual amount of TEA was determined by the difference in mass. It turned out that only in the case of using polyoxypropylene glycol as a medium, the mass fraction of TEA dissolved in it practically does not change, and triethylamine completely escapes from solutions in Puranol 373, polyoxytetramethylene glycol, and propylene glycol.

To explain the reasons for the retention of TEA by polyoxypropylene glycol, studies of the viscosity, density, and surface tension of TEA solutions in PPG were carried out. TEA solutions in PPG were also studied using ^1^H NMR and FTIR spectroscopy.

With an increase in the content of TEA in PPG (Figure 2), the dynamic viscosity of such a solution decreases many times over. The density of TEA solutions in PPG also decreases with an increase in the amount of TEA. Such a decrease in viscosity may be due to the significant effect of intermolecular interactions between PPG and TEA on the conformational response of polyoxypropylene glycol macrochains.

The ^1^H NMR spectra of TEOA/TEA solutions in PPG (Figure 3) show an upfield shift of the proton signals of the methyl, methylene, and methine groups relative to pure PPG. Since the shift of resonant signals of protons to a strong field is a consequence of an increase in their shielding, it can be concluded that the intermolecular interaction of PPG with TEOA/TEA leads to a change in the electron density distribution around the methyl, methylene, and methine groups of PPG.

FTIR spectroscopy was also used to study the features of the interaction of TEOA/TEA with PPG. The FTIR spectra show a shift in the absorption region of stretching vibrations of the C–H bond in the methylene group from 2866 cm^–1^ to 2870 cm^–1^ for the TEA solution in PPG relative to PPG (Figure 4b). As the content of TEA in PPG increases, the absorption bands at 2933 cm^–1^ are shifted due to stretching vibrations of C–H bonds in the methine group to longer wavelengths (Figure 4a). For methyl groups, on the contrary, in the analytical region of 2972 cm^–1^, the spectral absorption region does not change, but the absorption intensity increases in proportion to the increase in the TEA content in PPG (Figure 4a). A slight decrease in intensity is also observed for the analytical band at 1095 cm^–1^ due to stretching vibrations of the C–O bond in the PPG ether group (Figure 4c).

Intermolecular binding of PPG with TEOA leads to the formation of agglomerates characterized by a wide particle size distribution (Figure 5). Since the absorption intensity of the laser beam incident on the agglomerates is higher than the intensity of its reflection from the agglomerates, we can conclude that their macromolecular packing is loose. Agglomerates formed as a result of intermolecular binding of PPG and TEA are also characterized by a wide particle size distribution (Figure 5).

The fact that the intermolecular interactions of TEOA/TEA with PPG lead to the formation of certain supramolecular structures can also be judged from the isotherms of their surface tension, which differ markedly from the surface-active properties of individual PPG and TEOA/TEA. A low surface tension of such supramolecular structures and a pronounced dependence of the manifestation of surface tension isotherms on the content of TEOA (Figure 6a) and TEA (Figure 6b) in their composition with PPG are observed.

The totality of the results obtained makes it possible to explain the catalytic activity of triethylamine in the reaction of etherification of *ortho*-phosphoric acid by the hydrophobic-hydrophilic intermolecular interactions of TEA with PPG, which acquire a cooperative character. As a result of this process, supramolecular structures are formed, leading to a multiple decrease in the viscosity of the TEA solution in PPG and screening of TEA. The existence of TEA in the internal space of supramolecular formations prevents the direct interaction of TEA with *ortho*-phosphoric acid and the formation of ammonium phosphates but promotes the manifestation of its catalytic activity in the etherification reaction of *ortho*-phosphoric acid.

In the case of using triethanolamine, as a result of hydrophobic-hydrophilic interactions, shielding of TEOA molecules in the intermacromolecular space of polyoxypropylene glycol also occurs. Unlike triethylamine, the hydroxyl groups that make up TEOA enter into associative interactions with the etheric oxygen atoms of neighboring PPG chains, thus preventing the folding of PPG macrochains. As a result, the viscosity of the TEOA solution in PPG increases, in contrast to the viscosity of a similar TEA solution (Figure 2).

The chemical shifts of signals corresponding to methyl, methylene, and methine groups on the ^1^H NMR spectra of TEOA solutions in PPG are shifted similarly to TEA solutions in PPG to a higher field. When TEOA interacts with OPA and PPG, incomplete etherification of OPA is observed, while when using TEA, the etherification reaction goes further and leads to the formation of polyphosphates (Figure 1b).

A kinetic analysis of the OPA etherification reaction was carried out. As a variable parameter, acid-base titration was used to determine the conversion of the P-OH groups of *ortho*-phosphoric acid during the interaction of TEOA/TEA with OPA and PPG.

In order to confirm the proposed mechanism of catalysis by complexes of tertiary amines with PPG of the OPA etherification reaction and the activity of TEOA hydroxyl groups, a reaction system based on TEOA, OPA, and PPG was studied at a very low PPG content. In this case, the molar ratio [TEOA]:[OPA]:[PPG] = 1:1:0.5 was taken (Figure 7a). According to Figure 7 and the data given in Table 1, even at a very low content of the TEOA complex with PPG, triethanolamine enters into the etherification reaction of *ortho*-phosphoric acid. At the same time, the rate constant of the OPA etherification reaction with triethanolamine exceeds the rate constant of the OPA etherification reaction with polyoxypropylene glycol (Table 1).

At a molar ratio of [TEOA]:[OPA]:[PPG] = 1:1:0.5, 10 min after the start of the interaction, a viscous mass begins to precipitate, corresponding to the product of the interaction of TEOA with OPA. To study the processes occurring during the interaction of TEOA, OPA, and PPG, the FTIR spectra (Figure 8) of both the initial TEOA, OPA, and PPG and the products of their interaction in the region of stretching vibrations of C-O bonds in the C-O-C (1100 cm^−1^), P-O bonds in the P-O-C (980 cm^−1^), and P-O bonds in the P-OH groups (875 and 950 cm^−1^), reflecting the main structural features of these compounds. According to the FTIR spectra, the mother liquor is a polyoxypropylene glycol that has not reacted at the molar ratio of [TEOA]:[OPA]:[PPG] = 1:1:0.5. This is evidenced by the appearance on the spectra of only a band in the region of stretching vibrations of the ether group at 1100 cm^−1^. Spectrum 6 (Figure 8) does not show bands in the region of 980 cm^−1^ and 950 cm^−1^ due to stretching vibrations of P-O bonds in the structure of P-O-C and P-OH groups. Indeed, according to material balance calculations, PPG does not enter into OPA etherification when using the molar ratio [TEOA]:[OPA]:[PPG] = 1:1:0.5. The viscous mass has a spectrum different from the spectra of OPA and TEOA and the products of the interaction of OPA and TEOA carried out in the absence of any amount of PPG. The formation of P-O-C bonds in the structure of the product of the interaction of TEOA with OPA in the reaction system based on [TEOA]:[OPA]:[PPG] = 1:1:0.5 is evidenced by the presence of a band at 980 cm^−1^ in the FTIR spectra. According to the spectrum 4 in Figure 8, the interaction of TEOA and OPA leads to the formation of ammonium phosphate, the FTIR spectrum of which is largely similar to that of OPA.

The studies carried out allow us to assert that the direct action of triethanolamine on OPA at 90°C for several hours does not lead to the etherification of *ortho*-phosphoric acid by the hydroxyl groups of triethanolamine. The use of a small amount of PPG in the interaction of TEOA with OPA under the same reaction conditions leads to the etherification of *ortho*-phosphoric acid by triethanolamine. At the same time, the rate constant of the reaction of TEOA with OPA in the reaction system based on [TEOA]:[OPA]:[PPG] = 1:1:0.5 exceeds the rate constant of the reaction in the system based on [TEOA]:[OPA]:[PPG] = 1:(3÷9):6. The precipitation of a viscous mass and the absence of signs of PPG involvement in the reaction process in the system [TEOA]:[OFA]:[PPG] = 1:1:0.5 indicates that the hydroxyl groups of TEOA will enter into the etherification reaction of phosphoric acid first of all. That is, triethanolamine occupies a central position in the structure of AEPA-(3÷9).

In the case of a reaction system based on TEA, OPA, and PPG, the process proceeds in two stages (Figure 7b). The first stage most likely corresponds to the etherification reaction. This can also be judged by the similarity between the values of the etherification reaction constants for the systems [TEOA]:[OPA]:[PPG] = 1:3:3, [TEOA]:[OPA]:[PPG] = 1:6:6, and [TEA]:[OPA]:[PPG] = 1:3:6, [TEA]:[OPA]:[PPG] = 1:6:6 (Table 1).The second step of the kinetic curve corresponds to the involvement of OPA in the reaction of polyphosphate formation. Indeed, at a molar ratio of [TEA]:[OPA]:[PPG] = 1:9:6, such a large excess of OPA can be spent only on the formation of polyphosphates. Thus, the earlier analysis of the chemical structure of AEPA and EPA is consistent with the kinetic analysis.

According to Figure 9 and Figure 10, the reaction proceeds at relatively low temperatures as well. For research, AEPA-6 was synthesized. An increase in temperature leads to a regular increase in the rate constant of the OPA etherification reaction with polyoxypropylene glycol with the participation of TEOA as a co-reagent and catalyst (Figure 10). When T_synth._ = 100 °C is reached, a further increase in temperature is not accompanied by an increase in the reaction rate constant.

^31^P NMR spectroscopy corroborates the results of the kinetic analysis. According to Figure 11, the interaction of TEOA and OPA leads to formation of ammonium phosphate, which has the resonance signal at δP 0.4 ppm. The signal at 0.4 ppm does not appear in spectra of AEPA synthesized over a wide temperature range. It is also worth noting that even in the absence of tertiary amines, OPA is partially etherificated by polyoxypropylene glycol. However, even when the reaction of OPA with PPG was carried out for 2 h at T = 90 °C, the conversion of OPA remained low. In the spectrum of the reaction product of OPA with PPG, two broad signals at the chemical shifts of –0.15 and –0.27 ppm are also observed due to the different dissociation degree of OPA in this composition. Use of thriethanolamine leads to further etherification of the remaining OPA. An increase in the conversion level of OPA accompanying the growth of the synthesis temperature can be characterized by changes in the integral intensities (areas) of corresponding ^31^P NMR signals (Figure 12).

Synthesis at a higher temperature (T_synth._ = 110 °C) results in qualitative modifications of the structure of AEPA-6, which are produced under these conditions. The appearance of two distinct signals at 1.6–1.8 ppm reflects the fact that polyphosphate structures are produced together with OPA ethers when the synthesis temperature exceeds 100 °C.

### 3.2. AEPA-6-PU Films Characterization

The branched structure, the presence of spatially separated ion pairs, and terminal hydroxyl groups were prerequisites for the use of AEPA-6 for the synthesis of polyurethane ionomer films. To obtain polyurethane film materials, AEPA-6 was interacted with an aromatic polyisocyanate. The completion of the reaction of urethane formation was established using FTIR spectroscopy by the complete disappearance of the analytical band in the region of 2275 cm^–1^, which characterizes the isocyanate groups.

The fact that AEPA-6 was completely consumed when interacting with aromatic polyisocyanates was also judged by the results of gel-sol analysis of AEPA-6-PU, carried out using toluene as an extractant. The low values of the sol fraction, not exceeding 2.2 wt% for AEPA-6-PU, indicate the completion of the urethane formation reaction here.

Correspondingly, as the degree of conversion of OPA increases with increasing the synthesis temperature of AEPA-6, the content of ionogenic groups in the composition of AEPA-6-PU also increases. In addition, the content of unreacted OPA in AEPA-6 is significantly reduced. As a result, the best strength characteristics are observed for AEPA-6-PU based on AEPA-6 synthesized in the temperature range of 70–90 °C. (Figure 13). As the synthesis temperature of AEPA-6 increases to 100–110 °C, the strength characteristics of the corresponding AEPA-6-PU decrease noticeably. This may be due to an increase in the probability of reactions leading to the formation of polyphosphate structures in the composition of AEPA-6 with an increase in the synthesis temperature.

The structure of AEPA-6, determined by the temperature of its synthesis, also affects the thermomechanical behavior of the corresponding AEPA-6-PU films (Figure 14). The presence of unreacted OPA in the composition of AEPA-6 obtained at 40 °C leads to a noticeable plasticization of AEPA-6-PU. At the same time, the glass transition temperature of AEPA-6-PU is practically independent of the synthesis temperature of AEPA-6. With an increase in the AEPA-6 synthesis temperature to 110 °C, the polyphosphates formed under these conditions contribute to a decrease in the thermal stability of AEPA-6-PU.

## 4. Conclusions

The etherification reaction of *ortho*-phosphoric acid with polyoxypropylene glycol in the presence of tertiary amines is studied. The catalytic activity of TEOA and TEA in the etherification reaction of phosphoric acid is explained by hydrophobic-hydrophilic interactions of TA with PPG, leading, as a result of collective interactions, to a specific orientation of polyoxypropylene chains around the TA.

As a result, the viscosity of the TEA solution in PPG decreases many times over and the TEA shielding occurs. The existence of TEA in the internal space of the coiled macrochain of polyoxypropylene glycol prevents the direct interaction of TEA with *ortho*-phosphoric acid and the formation of ammonium phosphates but contributes to the manifestation of its catalytic activity in the etherification reaction of *ortho*-phosphoric acid.

In the case of using triethanolamine, the screening of TEOA molecules in the intermacromolecular space of polyoxypropylene glycol also occurs as a result of hydrophobic-hydrophilic interactions. Unlike triethylamine, the hydroxyl groups that make up TEOA enter into associative interactions with the etheric oxygen atoms of neighboring PPG chains, thus preventing the folding of PPG macrochains and contributing to an increase in the viscosity of the TEOA solution in PPG. As a result of the inclusion of TEOA in the composition of AEPA, as the etherification reaction of OPA with triethanolamine proceeds, the tertiary amine is removed from the shielding zone by polyoxypropylene glycol and, due to interaction with the remaining P-OH groups, passes into the composition of the tertiary ammonium ion. As a result, in the case of the TEOA with OPA and PPG interaction, incomplete etherification of OPA is observed. When using TEA, the etherification reaction goes further and leads to the formation of polyphosphates.

It was discovered that as the synthesis temperature rises, so does the rate of the etherification reaction. 

Based on AEPA-6 and PIC, polyurethane ionomer films were obtained. It was found that the structure of the AEPA-6, determined by the temperature of its synthesis, affects the thermomechanical behavior and the physico-mechanical properties of the corresponding AEPA-6-PU films.

## Data Availability

The data presented in this study are available on request from the corresponding author.

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
