# Peer review of "Catalytic Etherification of ortho-Phosphoric Acid for the Synthesis of Polyurethane Ionomer Films"

_polymers, 2022, doi:10.3390/polym14163295_

Round 1

Reviewer 1 Report

Dear Author,

The author tries to show the catalytic effect of triethylamine on the etherification of polyurethane-ionomers. Here are my general comments:

(1) What is the novelty of this work? similar type chemistry is well-known in literature.

(2) For boiling concerns, rather than triethylamine, other amines such as pyridine or Dimethylamine pyridine (DMAP) can be used.

(3) Does materials are well-purified? Can use dialysis to remove unreacted amine. Also, what is the yield of products? 

(4) Glass transition temperature, thermal, and mechanical properties are not explained. Interpretation of the data and how amine affects both properties should be included. 

(5) How FTIR can relate to the final product yield. Does it throw any light on unreacted starting materials? If yes, it should be explained in the main discussion to support your claim. 

(6) Authors also have not discussed about the phenomenon of work on urethane ionomers by timothy long's group. It should be mentioned in the introduction, how phosphorus-based are better.

Overall the manuscript needs more improvement.

Author Response

Dear Reviewer,

Thank you for your comments and feedback, which have helped us to substantially improve our manuscript. We carefully investigated you recommendations and prepared comments for each item you mentioned in your review.

(1) What is the novelty of this work? similar type chemistry is well-known in literature.

Answer: The literature review has been expanded and information about the novelty and relevance of this work has been supplemented.

(2) For boiling concerns, rather than triethylamine, other amines such as pyridine or Dimethylamine pyridine (DMAP) can be used.

Answer: The use of triethylamine as a model compound for studying the mechanism of the catalytic activity of tertiary amines in the OPA etherification reaction is due to the greatest similarity of its chemical structure with triethanolamine. The сorresponding changes were added to the experimental part. Line 174-176.

(3) Does materials are well-purified? Can use dialysis to remove unreacted amine. Also, what is the yield of products? 

Answer: The kinetic studies given in this manuscript have shown that triethanolamine containing terminal hydroxyl groups is completely consumed in the etherification reaction. Triethylamine is not involved in the formation of the structure of ethers of ortho-phosphoric acid, but is only a catalyst for the etherification reaction.

(4) Glass transition temperature, thermal, and mechanical properties are not explained. Interpretation of the data and how amine affects both properties should be included. 

Answer: A discussion of the features of the glass transition temperature is introduced. For Interpretation of the data and how amine affects mechanical properties, the corresponding addition is included in the introduction: «In [56], it was found that polyurethane ionomers obtained on the basis of AEPA, in the formation of the central unit of which triethanolamine participates, exhibit significantly higher mechanical properties compared to polyurethanes obtained on the basis of EPA.» Line 474-476.

(5) How FTIR can relate to the final product yield. Does it throw any light on unreacted starting materials? If yes, it should be explained in the main discussion to support your claim. 

Answer: FTIR spectroscopy was only used to investigate the interactions of TEOA/TEA with PPG. The corresponding addition was made to the manuscript: "FTIR spectroscopy was also used to study the features of the interaction of TEOA/TEA with PPG". 31P NMR spectroscopy was used to determine the yield of ortho-phosphoric acid etherification reaction products.

(6) Authors also have not discussed about the phenomenon of work on urethane ionomers by timothy long's group. It should be mentioned in the introduction, how phosphorus-based are better.

Answer: Discussion of works related to urethane ionomers by Timothy Long's group added to the introduction. Line 69-87 and 91-96.

Reviewer 2 Report

COMMENTS TO AUTHOR:

The paper entitled “Catalytic Etherification of ortho-Phosphoric Acid for the Syn- 2 thesis of Polyurethane Ionomer Films” by Ilsiya M. Davletbaeva and co authors comprehensively focus on the Polyurethane Ionomer Films.

I think it is of great interest in the community of Polyurethane Ionomer Films, because the high demand of Polyurethane Films. As a result, I will recommend the publication of this manuscript after major revision.

Comments:

1.    Page 3 line 92 Figure 1: Its hard to understand the reaction scheme because N showed valency of 3 and author have either mention wrong structure or missed positive sign on the NH.

2.    Figure 3 Kindly assign all the peaks on the NMR spectras and designate them on the structure respectively.

3.    Figure 8 Kindly assign all the frequencies properly on the FTIRs.

4.    Figure 11 Kindly assign the peaks and explain what extra peaks defines there.

5.    Kindly redraw some figures: the figures doesn’t look consistent.

Author Response

Dear Reviewer,

Thank you for your comments and feedback, which have helped us to substantially improve our manuscript. We carefully investigated you recommendations and prepared comments for each item you mentioned in your review.

  1. Page 3 line 92 Figure 1: Its hard to understand the reaction scheme because N showed valency of 3 and author have either mention wrong structure or missed positive sign on the NH.

Answer:Corrected. Figure 1 formation of AEPA. Line 117.

  1. Figure 3 Kindly assign all the peaks on the NMR spectras and designate them on the structure respectively.

Answer: Corrected. Figure 3 1H NMR spectra. Line 282.

  1. Figure 8 Kindly assign all the frequencies properly on the FTIRs.

Answer: Corrected. Figure 8 FTIR spectra. Additional clarifications have been added to the text. Line  368-373, 374-387, 385-387.

  1. Figure 11 Kindly assign the peaks and explain what extra peaks defines there.

Answer: Corrected.31P NMR spectra. Line 426.

  1. Kindly redraw some figures: the figures doesn’t look consistent.

Answer: Figures improved.

Round 2

Reviewer 1 Report

Dear Author,

The additional result and discussion help to understand the common readers. Paper can now be accepted.

Reviewer 2 Report

I think now the revised manuscript is in acceptable form.